# DC 4-Point Measurement for Total Electrical Conductivity of SOFC Cathode Material

**Kanghee Jo** [1,†], **Jooyeon Ha** [1,2,†], **Jiseung Ryu** [3], **Eunkyung Lee** [4] and **Heesoo Lee** [1,*]

1   School of Materials Science and Engineering, Pusan National University, Busan 46241, Korea; jokanghee@pusan.ac.kr (K.J.); hjyeon@ktl.re.kr (J.H.)
2   Material Technology Center, Korea Testing Laboratory, Seoul 08389, Korea
3   Analysis Technology Center, Korea Institute of Ceramic Engineering and Technology, Jinju 52851, Korea; jsryu@kicet.re.kr
4   Interdisciplinary Major of Maritime AI Convergence, Department of Ocean Advanced Materials Convergence Engineering, Korea Maritime and Ocean University, Busan 49112, Korea; elee@kmou.ac.kr
*   Correspondence: heesoo@pusan.ac.kr; Tel.: +82-51-510-2388
†   Shared 1st author.

**Abstract:** Conductive oxides are widely studied as cathode materials for electrochemical cells, such as solid oxide fuel cells (SOFCs), because of their chemical stability and high electrical conductivity at high temperatures (800–950 °C). The cathode is a key component of SOFCs, accounting for the greatest resistance loss among the SOFC components. It is important to precisely determine the conductivity of the cathode material, but it is difficult to achieve consistency among measurements because of errors caused by differences in the measurement methods and conditions employed by various research teams. In this study, the total electrical conductivity of an SOFC cathode material was measured by the DC 4-point method by investigating the geometrical parameters of the sample and the measurement terminal and the measurement device using $La_{0.8}Sr_{0.2}MnO_{3+d}$ (LSM). The measurement variables included the spacing between the measurement terminals (1 and 2 cm), lead wire diameter (0.25 and 0.5 mm), specimen thickness (3, 4, and 5 mm), and the applied current (10, 50, and 100 mA). The larger the spacing between the measurement terminal and the thinner the specimen, the smaller the standard deviation.

**Keywords:** solid oxide fuel cell cathode; conductive fine ceramics; LSM; DC 4-terminal method; total electrical conductivity

## 1. Introduction

A fuel cell is a high-efficiency, eco-friendly power generation device that electrochemically converts chemical energy into electrical energy. Solid oxide fuel cells (SOFCs) operate at high temperatures (800–1000 °C) because the cathode, anode, and electrolyte are all made of ceramic materials, and SOFCs exhibit fuel flexibility and a high efficiency of over 80% without the use of noble metal catalysts such as Pt and Au [1–3]. As the demand for eco-friendly, high-efficiency electrochemical systems such as water electrolysis cells and metal smelting processes continue to increase, the application of conductive ceramic materials is expected to expand [4,5].

The efficiency of SOFCs is known to be influenced by the electronic conductivity of the cathode and anode and the oxygen ionic conductivity of the electrolyte [6,7]. The cathode material is essential in SOFC systems and can affect the output of the system [8,9]. Among the various characteristics of cathode materials for SOFCs, the total electrical conductivity plays an important role in enabling the oxygen reduction reaction (ORR) electrochemical catalyst to exhibit catalytic activity over the entire area of the material, and it is known that the conductivity of the cathode material is preferred to be more than 100S/cm. [10,11]. Accordingly, it is important to accurately and precisely measure the electrical conductivity of a material.

Methods of measuring the total electrical conductivity of conductive oxide materials include the DC 2-point method and the DC 4-point method. In the DC 2-point method (Figure 1a), current flows through two electrodes connected to the specimen, and the resistance of the specimen is also quantified through a voltmeter-connected electrode parallel to the circuit. The measured value is not an accurate reflection of the true value because of the contact resistance caused by the oxide film on the specimen surface and the resistance of the electrode itself. On the other hand, in the case of the DC 4-point method, the current flowing into the measuring electrode flows on the pA level, so that measurement errors can be greatly reduced because the electrodes that pass current through the specimen and the electrodes that measure voltage are separated, as shown in Figure 1b [12]. The obtained data may not be consistent for various research teams owing to differences in the measurement conditions, etc. Additionally, it is known that the conductivity of the materials is affected by the fabrication method [13,14]. Different fabrication methods result in different grain sizes that affect the contributions of the grains and grain boundaries [13], as well as in the presence or absence of undesirable secondary phases [14].

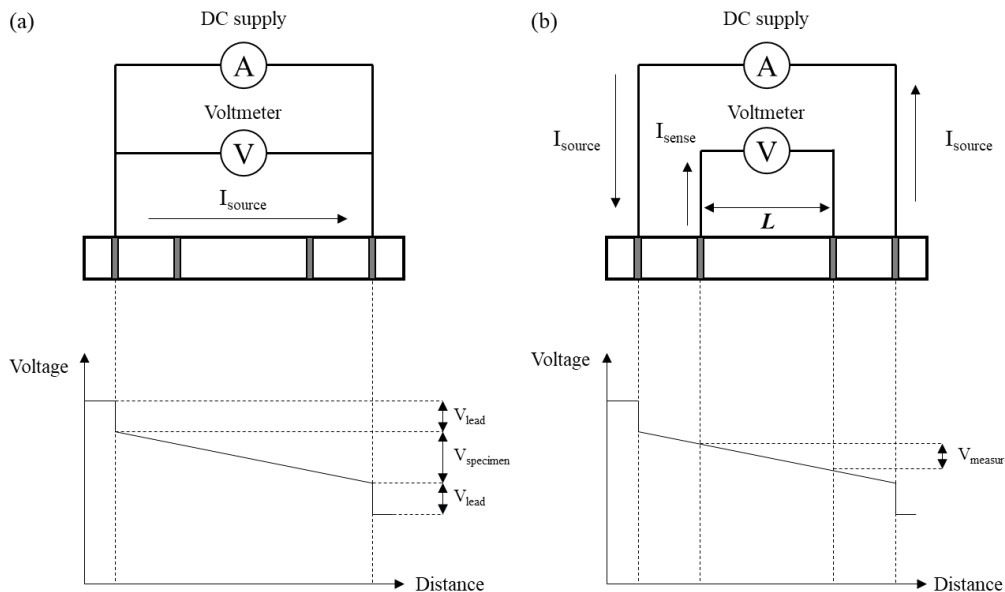

**Figure 1.** The comparison between DC 2-point method (**a**) and DC 4-point method (**b**).

If the conductivity measurement method using the DC 4 point-probe method for conductive ceramics, including SOFC cathode materials, is standardized, data reliability and compatibility between research groups can be improved. Herein, we investigate and optimize various measurement conditions and techniques for the DC 4-electrode measurement method to obtain precise and accurate electrical conductivity data for conductive ceramics. The source of the error for each condition is identified.

## 2. Materials and Methods

Two types of $La_{0.8}Sr_{0.2}MnO_{3+\delta}$ powder (LSM82-N, K-ceracell, Korea and LSM20-HP, Fuelcellmaterials, USA) were uniaxially pressed at a pressure of 144.92 MPa to form a bar-type green body. The green bodies were sintered at 1400 °C for 4 h to obtain a $3 \times 5.5 \times 40$ mm$^3$ specimen, which was ground to dimensions of $3 \times t \times 40$ mm$^3$ (where t is the thickness, Figure 2) using a low-speed cutting tool (Minitom, Struers). The surface of the ground specimen was polished with #1200 and #1500 sandpaper; the measurement conditions are shown in Table 1.

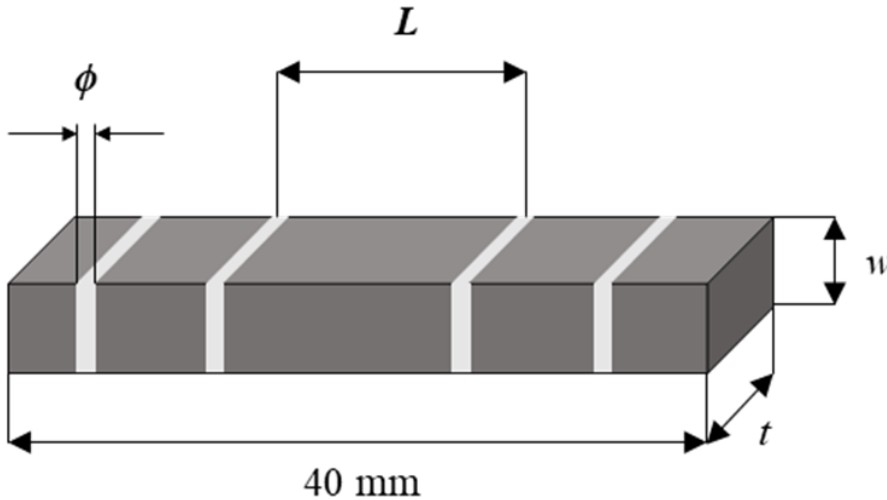

**Figure 2.** Schematic diagram of specimen.

**Table 1.** Experiment variables of DC 4-point method.

| Variables | Dimension |
|---|---|
| Width ($w$) | 3 mm |
| Sample thickness ($t$) | 3, 4, 5 mm |
| Spacing between the measurement terminals ($L$) | 10, 20 mm |
| Diameter of lead wire ($\phi$) | 0.25, 0.50 mm |
| Applied current ($I$) | 10, 50, 100 mA |

The crystal structures of the two powders used in the experiment were measured at intervals of $0.02°/2\theta$ in the $2\theta$ range of 10–90° using powder X-ray diffraction (XRD PANalytical X'pert-Pro MPD PW3040/60, Malvern Panalytical, Almelo, Netherlands). The surface morphology of the polished specimens was confirmed using field-emission scanning electron microscopy (FE-SEM, Hitachi S-4800, Hitachi, Tokyo, Japan).

To measure the overall conductivity, two Pt wires were fixed around the edge of the specimen, and the other two were fixed around the center of the specimen. To calculate the spacing (L) of the inner Pt wire and the cross-sectional area (A) of the specimen, the width and height were measured using a Vernier caliper. The specimen was fixed in a tube furnace, heated at a rate of 10 °C min$^{-1}$ to the desired temperature, and maintained for approximately 10 min. Finally, the two outer wires were connected to a DC power supply (Keithley 2400) to pass a constant current, and the two inner wires were connected to a digital multimeter (Agilent 34401A digital multimeter) to measure the potential. The measurement conditions are listed in Table 1. Using the dimensions of the test specimen, the applied current, and the measured voltage, the total conductivity of the specimen was calculated using Equation (1) (Where, $\sigma$ is the conductivity of the specimen (S/cm), $V$ is the measured voltage (V), and A is the cross sectional area of the specimen (cm$^2$); the other variables are described in Table 1 [15].

$$\sigma = \frac{L}{R \times w \times t} = \frac{I \times L}{V \times A} \qquad (1)$$

## 3. Results

Figure 3 shows the X-ray diffraction pattern of the sintered specimens of LSM_F and LSM_K. For the XRD patterns of LSM_F and LSM_K in Figure 3, all peaks were indexed to the La$_{0.8}$Sr$_{0.2}$MnO$_3$ (LSM) perovskite structure, and no secondary phase was observed. Both LSM_F and LSM_K had the same crystal structure as La$_{0.8}$Sr$_{0.2}$MnO$_3$ (ICSD 98-005-

1655), with the R-3m hexagonal structure. Figure 4a,b shows the surface morphology of the sintered body. Both specimens had a density and porosity of about 6.4g/cm$^3$ and 0.09%, respectively. The grain sizes were calculated as 3.29 μm for LSM_K and 2.74 μm for LSM_F, according to the Scherrer equation, which corresponded to SEM image. Therefore, it is expected that the difference between the measured values for the two samples will be minimized. Factors that can affect the DC 4-point measurement method are the shape of the specimen, the distribution of defects within the specimen, the contact between the lead wire and the specimen, and the offset of the measuring device, having excluded the characteristics related to the crystal structure and microstructure of the sample.

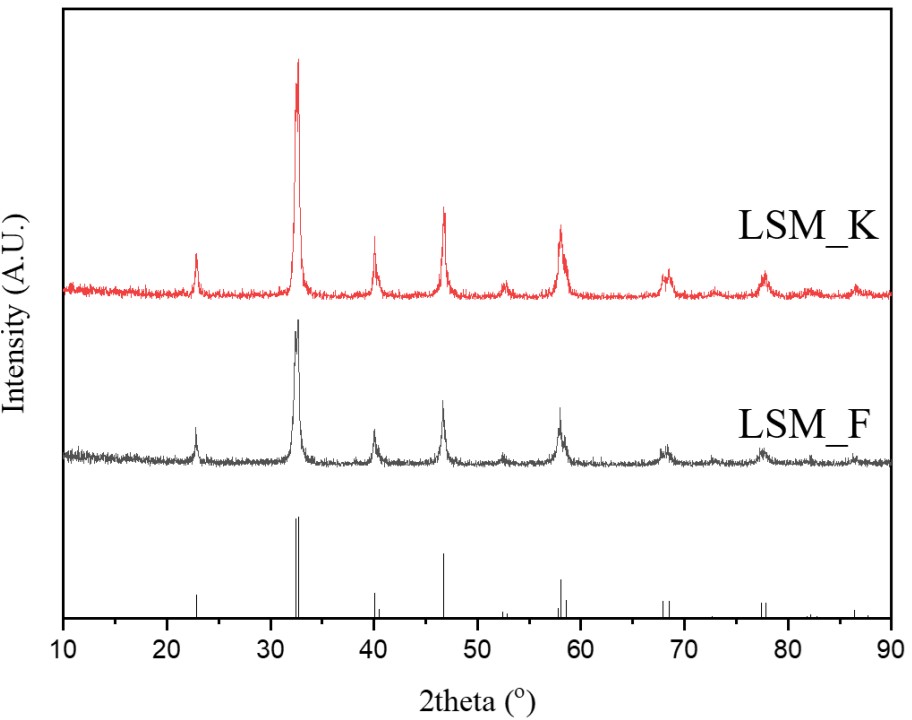

**Figure 3.** XRD patterns of LSM powders.

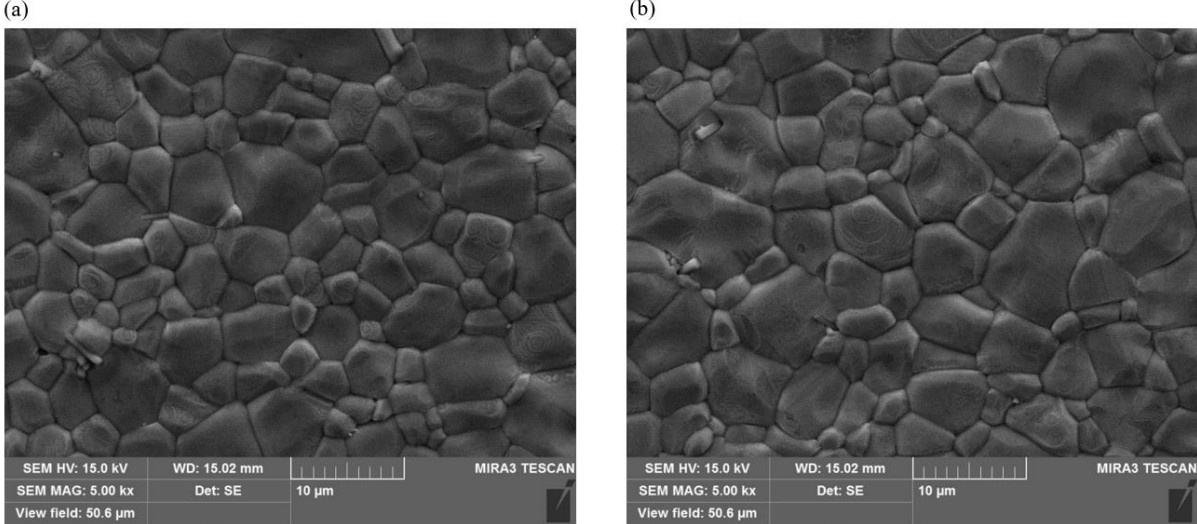

**Figure 4.** Surface morphology of sintered sample: (**a**) LSM-F and (**b**) LSM-K.

The effect of the spacing between the lead wires on the conductivity was evaluated, as shown in Figure 5. The conductivity of each specimen with spacings of 1 cm and 2 cm were calculated after fixing the specimen thickness (5 mm), lead wire diameter (0.50 mm), and applied current (50 mA). Both specimens showed a value of approximately 200 S/cm at 900 °C, which is similar to that of LSM, as shown in Table 2. In the case of LSM_F, the average and standard deviation value at 900 °C, which is the operating temperature of LSM, was 187.34 ± 20.31 S/cm for the 1 cm electrode spacing and 210.54 ± 11.28 S/cm for the 2 cm electrode spacing. For LSM_K, the average and standard deviation value was 201.97 ± 13.74 S/cm at 1 cm and 204.85 ± 8.00 S/cm at 2 cm, and both specimens had a smaller standard deviation of conductivity at 2 cm. It is considered that a wider spacing of the lead wires is advantageous for more evenly included defects such as grain boundaries and pores, which are randomly distributed in the specimen.

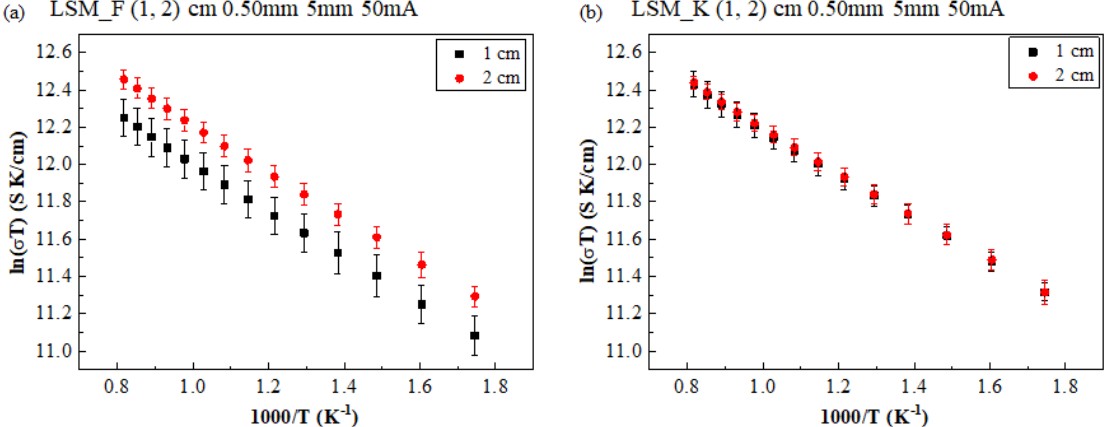

**Figure 5.** Arrhenius plot of measured electrical conductivity according to lead wire spacing: (**a**) LSM-F and (**b**) LSM-K.

**Table 2.** Literature review on Electrical conductivity of LSM.

| Material | Electrical Conductivity at 900 °C (S/cm) | Fabrication Method | Sintering Condition (Temperature, Hour) | Reference |
|---|---|---|---|---|
| LSM82 | 209 | Acetate aqueous solution | 1400 °C | [16] |
| LSM73 | 235 | Commercial | 1450 °C, 5 h | [17] |
| LSM82 | 270 | Commercial | 1300 °C, 5 h | [18] |
| LSM82 | 229 (800 °C, Van der Pauw) | Sol-gel method | 1050 °C, 2 h (Brush coating) | [19] |
| LSM82 | 190 | ECCP | 1250 °C, 4 h | [20] |

To evaluate the effect of the contact between the specimen and the lead wire, the conductivity of each specimen according to the diameter of the lead wire was determined, as shown in Figure 6, where the specimen thickness (5 mm), the lead wire spacing (2 cm), and the applied current (50 mA) were fixed. When the thickness of the lead wire in LSM_F was 0.50 mm, the average and standard deviation value was 210.54 ± 11.28 S/cm, and for a thickness of 0.25 mm, the average and standard deviation value was 207.86 ± 12.97 S/cm. For both specimens, the standard deviation of the conductivity was smaller for the lead wire thickness of 0.50 mm. When the lead wire is thick, there is a smaller deviation because the lead wire of 0.50 mm can make more stable contact with the sample than the 0.25 mm counterpart.

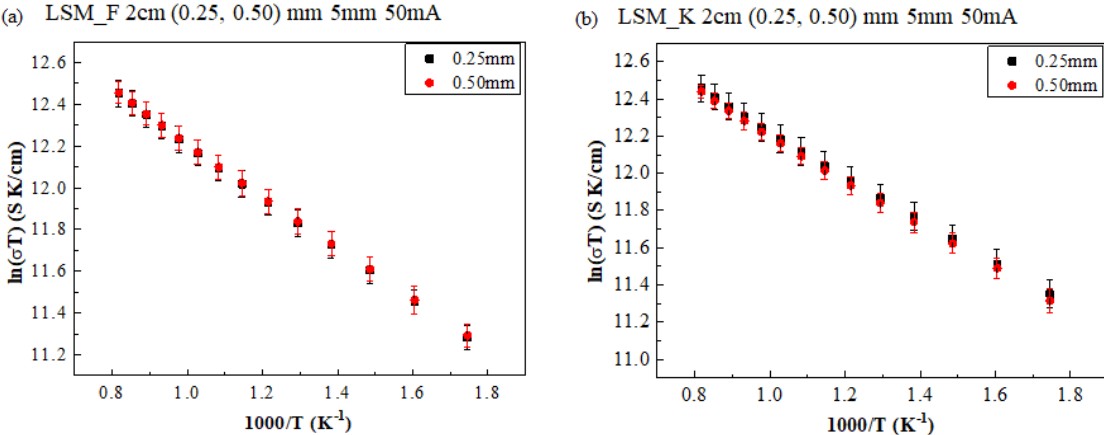

**Figure 6.** Arrhenius plot of measured electrical conductivity according to lead wire diameter: (**a**) LSM-F and (**b**) LSM-K.

To evaluate the effect of the cross-sectional area of the specimen, the conductivity was measured by changing the thickness of the specimen to 5 mm, 4 mm, and 3 mm with a lead wire spacing of 2 cm, lead wire diameter of 0.50 mm, and applied current of 50 mA (Figure 7). For LSM_F and LSM_K, the standard deviation gradually decreased from 11.21 S/cm and 8.37 S/cm to 3.36 S/cm and 0.35 S/cm, respectively. In both cases, the conductivity had a smaller standard deviation as the cross-sectional area of the sample decreased. It was determined that the deviation of the measured value decreased as the cross-sectional area became thinner due to the decreased number of defects in the sample after passing through the grinding and polishing process.

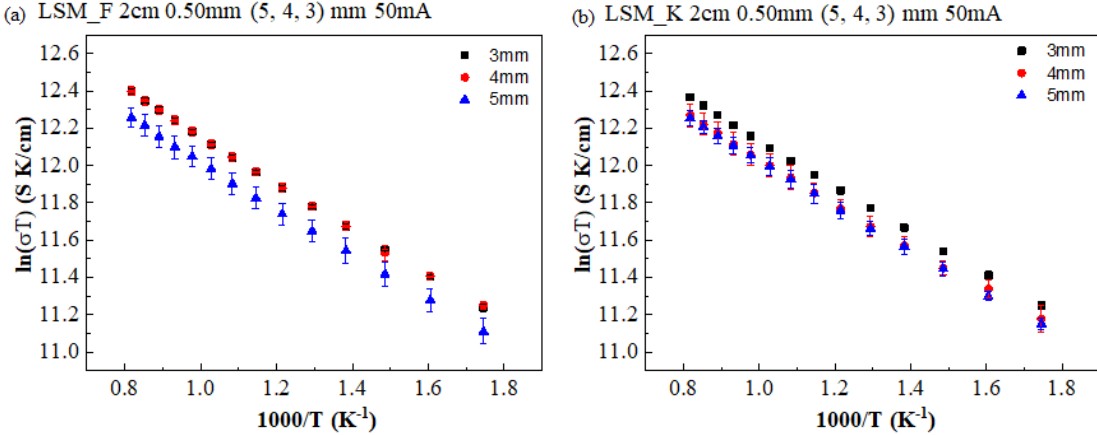

**Figure 7.** Arrhenius plot of measured electrical conductivity according to sample thickness: (**a**) LSM-F and (**b**) LSM-K.

The geometric parameters of the specimen and lead wire were fixed by the specimen thickness of 3 mm, lead wire spacing of 2 cm, and lead wire diameter of 0.5 mm. The conductivity was measured to consider the effect of the measurement device by varying the applied current (10, 50, and 100 mA, Figure 8). For an applied current of 10 mA, the average and standard deviation value in the conductivity of LSM_F and LSM_K was 206.11 ± 4.78 S/cm and 192.76 ± 9.92 S/cm, respectively. When the applied current was increased to 100 mA, the standard deviation in the conductivity of LSM_F and LSM_K decreased 2.35 S/cm and 0.39 S/cm, respectively. This is because the influence of the offset value when the current is not applied gradually decreases as the applied current increases. Based on these results, more precise and accurate measurement values can be obtained when using the 4-electrode method to investigate the conductivity of a conductive

ceramic bulk sample by employing a longer distance between the lead wires for the voltage measurement, thinner sample, and larger applied current.

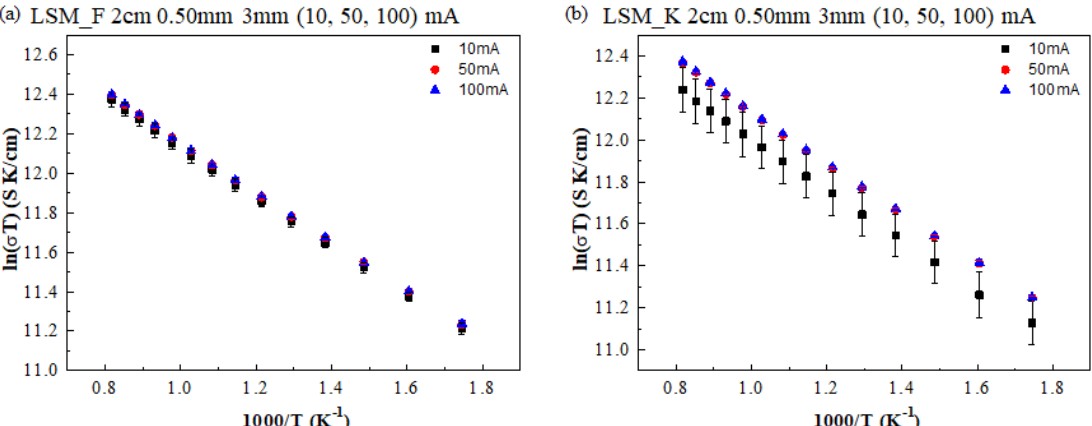

**Figure 8.** Arrhenius plot of measured electrical conductivity according to applied current: (**a**) LSM-F and (**b**) LSM-K.

## 4. Conclusions

In this study, the DC 4-point method using La0.8Sr0.2MnO3+d (LSM) was optimized in terms of the geometrical relationship between the sample and the lead wire used for measurement and the measurement device. The control variables were the distance of the lead wire (1 cm, 2 cm), the diameter of the lead wire (0.25, 0.5 mm), the thickness of the specimen (3, 4, 5 mm), and the applied current (10, 50, 100 mA). When the lead wire diameter was varied, the standard deviation was 11.21 S/cm at 0.5 mm and 13.06 S/cm at 0.25 mm. Compared to other variables, it was determined that the diameter of the lead wire did not significantly affect the precision of the experiment. When the spacing between the lead wires was 2 cm and the specimen thickness was 3 mm, the smallest standard deviation (0.35 S/cm) was observed, which means that the experimental conditions that minimize defects in the sample and include all the defects are more advantageous experimental conditions. In addition, the larger the applied current, the smaller the deviation, which was found to be less affected by the offset of the device.

**Author Contributions:** Conceptualization, K.J., J.R. and H.L.; methodology, J.H. and J.R.; validation, K.J. and J.H.; formal analysis, K.J.; investigation, E.L.; resources, K.J.; writing—original draft preparation, K.J. and J.H.; writing—review and editing, J.R. and H.L.; visualization, J.H.; supervision, H.L.; project administration, H.L.; funding acquisition, H.L. All authors have read and agreed to the published version of the manuscript.

**Funding:** This work was supported by the Korea Institute for Advancement of Technology grant funded by the Korea Government (MOTIE) (P0008335, The Competency Development Program for Industry specialist).

**Institutional Review Board Statement:** Not applicable.

**Informed Consent Statement:** Not applicable.

**Data Availability Statement:** The data presented in this study are available on request from the corresponding author.

**Conflicts of Interest:** The authors declare no conflict of interest.

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
