# Peer review of "DC 4-Point Measurement for Total Electrical Conductivity of SOFC Cathode Material"

_applsci, doi:10.3390/app11114963_

Round 1

Reviewer 1 Report

Revision of MS-1217830, “DC 4-Point Measurement for Total Electrical Conductivity of SOFC Cathode Material”.  Submitted in MDPI Applied Sciences

Dear Editor,

This manuscript is very interesting to those who work in the field of Solid Oxide Fuel Cells. But I feel the manuscript needs a lot of work (major revision) to improve enough to merit its publication.

Introductory section.

The manuscript needs more comments on the materials employed, their conductivities… (and include more references). I suggest the authors to check in the literature for review manuscripts were they can discuss on the conductivities of materials. This should be added in the first paragraph.

[REF_A] Mahato, N., Banerjee, A., Gupta, A., Omar, S., Balani, K. Progress in material selection for solid oxide fuel cell technology: A review. 2015. Progress in Materials Science, 72, 141-337. https://doi.org/10.1016/j.pmatsci.2015.01.001

 [REF_B] Sun, C., Hui, R. & Roller, J. Cathode materials for solid oxide fuel cells: a review. J Solid State Electrochem 14, 1125–1144 (2010). https://doi.org/10.1007/s10008-009-0932-0

[REF_C]H. A. Taroco, J. A. F. Santos, R. Z. Domingues and T. Matencio (August 9th 2011). Ceramic Materials for Solid Oxide Fuel Cells, Advances in Ceramics - Synthesis and Characterization, Processing and Specific Applications, Costas Sikalidis, IntechOpen, DOI: 10.5772/18297. Available from: https://www.intechopen.com/books/advances-in-ceramics-synthesis-and-characterization-processing-and-specific-applications/ceramic-materials-for-solid-oxide-fuel-cells

[REF_D] Zakaria, Z., Mat Z.A., Hassan, S.H.A., Kar, Y.B. A review of solid oxide fuel cell component fabrication methods toward lowering temperature. 2020. Int. J. Energy Res., 44, 594-611. https://doi.org/10.1002/er.4907

 [REF_E] Hussain, S., Yangping, L. Review of solid oxide fuel cell materials: cathode, anode, and electrolyte. 2020. Energy Transit 4, 113–126. https://doi.org/10.1007/s41825-020-00029-8

[REF_F] Zhang, J., Lenser, C., Menzler, N. H., Guillon, O. Comparison of solid oxide fuel cell (SOFC) electrolyte materials for operation at 500 °C. 2020. Solid State Ionics, 344, 115138. https://doi.org/10.1016/j.ssi.2019.115138.

Please add in the text the following sentence (or similar), together with the following references: Also, it is known that conductivity of the materials is affected by the fabrication method [REF_G REF_H]. Fabrication method yields in different grain sizes that affect the contributions for the grains and grain boundaries [REF_G], as well as in the presence or absence of undesirable secondary phases [REF_H].

[REF_F] Garcia-Garcia, F.J., Tang, Y., Gotor, F.J., Sayagués, M.J. Development by Mechanochemistry of La0.8Sr0.2Ga0.8Mg0.2O2.8 Electrolyte for SOFCs. 2020, Materials, 13, 1366.

[REF_G] Garcia-Garcia, F.J., Sayagués, M.J., Gotor, F.J.  A Novel, Simple and Highly Efficient Route to Obtain PrBaMn2O5+δ Double Perovskite: Mechanochemical Synthesis. 2021. Nanomaterials,  11(2), 380. https://www.mdpi.com/2079-4991/11/2/380

Materials and Methods.

Authors mention that “were uniaxially pressed at a pressure of 20 MPa to form a bar-type green body”.  However, typical pressures applied for the compression on these materials are in the order of 100 MPa. Please comment on this issue or provide density values for final specimens.

They also claim that “The green bodies were sintered at 1400 °C for 4 h”, but to me 4h is very short time. Please compare with values published in literature, because I remember typical sintering times are around 10 hours.

Results section.

Authors mention in page 3 that “Both specimens were sintered without pores”. But this is not scientific terminology (authors cannot observe the presence of nano- and mesopores occluded, more if it is taken into account that they employed 20 MPa). Please provide values for density measured by Archimedes, which is a very precise and well-stablished method. Authors can use another method if they prefer, but density values should be provided.

Figure 3: Please label the diffraction maxima and provide in the text the crystalline domain size by the Scherrer method, for a better comparison of the 2 specimens.

Authors claim in page 4 that “200 S/cm at 900 ºC is similar to that of LSM reported in the literature [10]”. However, LSM is being measured on many occasions, and this value depends on different factors, such us grain size, crystalline domains, fabrication method… For a reasonable comparison of values obtained and values available in literature, I honestly recommend that authors add a table comprising, fabrications methods (or any other distinguishable property), conductivities and references.  

With Best Regards,

The Reviewer.

Round 2

Reviewer 1 Report

Dear Editor,

I see authors made a great effort amended this manuscript. The amended manuscript has improved considerably and deserved to be published in its current form.

Reviewer 2 Report

The article deserves publication